# P84/ZCC Hollow Fiber Mixed Matrix Membrane with PDMS Coating to Enhance Air Separation Performance

**DOI:** 10.3390/membranes10100267

**Published:** 2020-09-28

**Authors:** Nurul Widiastuti, Triyanda Gunawan, Hamzah Fansuri, Wan Norharyati Wan Salleh, Ahmad Fauzi Ismail, Norazlianie Sazali

**Affiliations:** 1Department of Chemistry, Institut Teknologi Sepuluh Nopember, Faculty of Science and Data Analytics, Sukolilo 60111, Surabaya, Indonesia; triyanda@its.ac.id (T.G.); h.fansuri@chem.its.ac.id (H.F.); 2Advanced Membrane Technology Research Centre (AMTEC), Universiti Teknologi Malaysia, Skudai 81310, Johor Bahru, Malaysia; w-norharyati@utm.my (W.N.W.S.); afauzi@utm.my (A.F.I.); 3Centre of Excellence for Advanced Research in Fluid Flow (CARIFF), Universiti Malaysia Pahang, Lebuhraya Tun Razak, Gambang 26300, Pahang, Malaysia; azlianie@ump.edu.my

**Keywords:** zeolite carbon composite, mixed matrix membrane, PDMS coating, P84 co-polyimide, air separation

## Abstract

This research introduces zeolite carbon composite (ZCC) as a new filler on polymeric membranes based on the BTDA-TDI/MDI (P84) co-polyimide for the air separation process. The separation performance was further improved by a polydimethylsiloxane (PDMS) coating to cover up the surface defect. The incorporation of 1 wt% ZCC into P84 co-polyimide matrix enhanced the O_2_ permeability from 7.12 to 18.90 Barrer (2.65 times) and the O_2_/N_2_ selectivity from 4.11 to 4.92 Barrer (19.71% improvement). The PDMS coating on the membrane further improved the O_2_/N_2_ selectivity by up to 60%. The results showed that the incorporation of ZCC and PDMS coating onto the P84 co-polyimide membrane was able to increase the overall air separation performance.

## 1. Introduction

The development of gas separation by membrane technology has increased rapidly in the last two decades. The technology offers ease of operation, low energy requirement, and an overall economy of low-scale operation [1]. Polymeric membranes are the most used materials for gas separation membranes. However, they have some limitations, such as poor thermal and low chemical resistance [2]. Choosing a proper polymer precursor is necessary to avoid these drawbacks. Polyimides are one of the most preferred polymers used in the membrane community [3,4,5,6,7,8,9]. They are polymers with good thermal properties (Tg~300 °C). Furthermore, they are also low cost, have high mechanical properties, and can be modified to different configurations (flat, hollow fiber, and supported membranes). In addition, these polymers belong to a family of precursors that are optimal for the preparation of carbon molecular sieve membranes [2,10]. However, as other polymeric membranes, this membrane suffers from the trade-off between selectivity and permeability, known as “upper-bound”, as proposed by Robeson, who stated that “the gas separation properties of the polymeric membranes follow distinct trade-off relations, where more selective membranes are generally less permeable and vice versa” [11,12,13].

Advanced membrane preparation methods, such as polymer blends and their composites, are new promising methods due to their ability to enhance the thermal, mechanical, and separation properties of the membrane. Compositing a polymer with an inorganic filler is the most efficient way to improve the overall membrane performance [14]. The method continues to receive a lot of interest because it offers significant performance improvement of membranes, such as good separation factor, mechanical strength, thermal and chemical stability [15]. Many types of fillers have already been reported, such as zeolite [16], silica [17], metal-organic framework (MOF) [18] and carbon molecular sieve (CMS) [19]. Among them, zeolite exhibits interesting physicochemical properties, namely sorption and diffusion, in gas separation applications. These characteristics prevail due to the presence of channels and cavities of different sizes that are related to the void volume and free space, respectively [20]. Numerous studies on the use of zeolite as a filler in the polymer matrix were conducted [21,22,23,24], with notable selectivity improvement by 2 to 10 fold. However, a significant improvement in selectivity was accompanied with a drastic decline in permeation. The problem that usually appears when incorporating zeolite into the polymer matrix is the void formation, which leads to a performance slump due to incompatibility. Poor adhesion at the zeolite–polymer interface could lead to the “sieve-in-cage” morphology that is responsible for reduced separation performance [20]. Fortunately, the incompatibility issues can be fixed by utilizing light surface modification using suitable chemicals, such as silane and carbon groups.

In this study, the zeolite–carbon composite was used as a filler material into BTDA-TDI/MDI (P84) co-polyimide to form a mixed matrix membrane. A modification in the zeolite pore by carbon coating through the impregnation of sucrose as a carbon precursor was conducted. The modification was inspired by the fact that polymeric membranes are mostly operated at ambient conditions. Under these conditions, the presence of abundant moisture can lead to the formation of hydrogen bonding between water molecules in the moisture and oxygen species in the Si–O–Al frameworks of the zeolite structure [25,26,27], which reduces the permeation rate of the permeating gas. In addition, high zeolite loading leads to the formation of interfacial gaps that cause an increase in the membrane brittleness and a loss of selectivity [28]. Therefore, in this research, the zeolite pore wall was coated with a less hydrophilic material without compromising the permeation rate of penetrated gas. Carbon is a less hydrophilic material that has the potential to be impregnated as a precursor into zeolite pores to produce the zeolite carbon composite (ZCC). In the zeolite-based mixed matrix membranes, adding more filler usually causes the formation of more voids at the polymer–filler interface, resulting in the loss of selectivity. In addition, when a large amount of filler is added to the polymer matrix, the material becomes more brittle [24]. Thus, small amounts of fillers are preferred to avoid problems that were found when zeolite was used as the filler, as well as to reduce the productional cost of the membrane.

Previously, ZCC could be obtained during zeolite-templated carbon synthesis (ZTC) [29,30]. ZCC is the preceding material obtained prior to the template removal of ZTC using strong acids such as sulfuric acid and hydrofluoric acid. Unlike other zeolite–carbon, this composite is literally a zeolite where its pores are filled by carbon material, and some of its surfaces are also covered by a thin carbon layer. Few literatures discuss ZCC in more detail than ZTC because ZCC has a very small surface area to be applied as an adsorbent. However, this material has high potential as a suitable membrane filler due to its high micropore content and the presence of zeolite, and because it uses no hazardous chemicals in the preparation, such as hydrofluoric acid, in the preparation of ZTC. The zeolite used in this research was zeolite-Y (NaY), which is widely known in gas-related applications, while sugar was used as a carbon precursor due to its high carbon content and wide availability [29,30,31,32]. The motivation to utilize ZCC as a new filler for polymer membrane was due to the high microporosity that could improve the gas selectivity and pore regularity in order to improve permeability and solve the compatibility issue with the polymer.

The main challenge in fabricating a mixed matrix membrane is the formation of defects on the membrane surface. Introduction of hard-inorganic filler would result in the formation of void gaps between the polymer and filler interface, which lead to selectivity loss [33]. Thus, it is essential to seal those defects. An established technique to solve the issue is using polydimethylsiloxane (PDMS) to coat the membrane surface. However, the PDMS coating is usually accompanied with a reduction in permeability. Therefore, it is essential to study the effectiveness of the PDMS coating on the produced membrane. In this study, the produced hollow fiber membrane was coated with PDMS. The membrane was characterized by X-ray diffraction (XRD), scanning electron microscopy (SEM), Fourier transform infrared spectroscopy (FTIR) and thermogravimetric analysis (TGA) to investigate the properties of the membrane as well as the membrane separation performance on O_2_/N_2_ separation.

## 2. Materials and Methods

### 2.1. Material

Zeolite-Y was prepared using sodium aluminate (NaAlO_2_, Sigma-Aldrich) as an aluminum source for zeolite-Y synthesis, while the silicate source was sodium silicate solution (Na_2_SiO_3_, Sigma-Aldrich, St. Louis, MO, USA). The additional sodium counter ion was provided by sodium hydroxide (99% NaOH, pellet, Sigma-Aldrich, St. Louis, MO, USA). Sucrose (98%, Fluka, Singapore, Singapore) was used as carbon precursor to prepare ZCC filler.

The raw materials for membrane preparation were P84 co-polyimide (BTDA-TDI/MDI, HP Polymer, Lenzing, Austria) as polymer precursor and N-methyl-2-pyrrolidone (NMP, Merck, Darmstadt, Germany) as solvent. Polydimethylsiloxane (PDMS, Sylgard^®^ 184, Midland, MI, USA) as membrane surface coating was purchased from Dow Corning. PDMS coating solution was prepared by dissolving PDMS in n-hexane (Sigma-Aldrich, St. Louis, MO, USA).

### 2.2. Membrane Preparation

The filler, ZCC, was synthesized using the impregnation method of sucrose into zeolite-Y pores with a mass ratio of 1:1.25, followed by carbonization at 800 °C in nitrogen atmosphere. More detailed methods on ZCC synthesis were described elsewhere [29]. The hollow fiber membrane was fabricated via dry/wet spinning method with the parameters inspired from Favvas et al. [34,35] and Choi et al. [34,35] with some adjustments. The P84 co-polyimide and ZCC were dehydrated at 80 °C for 24 h to remove the moisture in advance of membrane preparation. For the mixed matrix membrane, the ZCC 1 wt% (0.2 g) was first dispersed in the NMP solvent using a sonicator (Qsonica, duration = 1 min, amplitude = 70%, pulse on and off = 10s) several times until fully dispersed in the solvent. The P84 co-polyimide powder was then added into solution slowly while being stirred mechanically at 700 rpm and 80 °C [36]. The dope solution that consisted of P84/NMP/ZCC (20/80/0-1 *w*/*w*) and bore liquid (70/30 *w*/*v* of NMP/H_2_O) was pumped at once through a tube-in-orifice spinneret using gear pumps. The inner diameter (i.d.) and outer diameter (o.d.) of the spinneret were 400 and 800 µm, respectively. The extruded fibers led through a 5 cm air gap prior to inflowing the room temperature coagulation bath, which was filled with tap water. The excess of NMP was further removed through ethanol immersion for 2 h and was allowed to dry at room temperature. The PDMS coating method was based on our previously reported study [36]. The prepared membrane then was referred to as a mixed matrix membrane (MMM).

### 2.3. Sample Characterizations

An X-ray diffractogram (XRD) was employed to confirm the structure formation of zeolite Y, ZCC and MMM. The sample morphology was monitored using a scanning electron microscope (SEM) (Hitachi, TM 3000, Tokyo, Japan) and the results were analyzed using ImageJ software. The accelerating voltage in SEM analysis was 15 kV, and the sample was coated with platinum. Fourier transform infrared spectroscopy (FTIR, Thermo Scientific Nicolet iS10, Waltham, MA, USA) was employed to observe the alteration in the functional group on the membrane. The thermal stability and glass transition of the membrane was analyzed using a thermal gravimetric analyzer (TGA, TA instrument TGA Q500, New Castle, DE, USA).

### 2.4. Pure Gas Measurement

For the gas measurement, in all case, five fibers (~15 cm in length) were assembled in a lab-scale module. The fibers were potted in a Swagelok 3164B3 NSNP, and the permeation performance was evaluated in a custom-made high-pressure gas permeation hollow fiber rig (1/4 in stainless steel (SS) 316 tubes) connected directly to a bubble meter. The single gas permeability was conducted at room temperature (~25 °C) and 4 bar of pressure. The gas volume was measured using a bubble flow meter. The measurement was in triplicate, and the results were presented as the average of three measurements. The permeability was estimated using Equation (1).
(1)$$P_{i}=\left(\frac{Q\;\times \;l}{\Delta P\;\times \;A}\;\right)=\frac{Q\cdot l}{n\pi D\Delta P}$$
where *Q* is the volumetric gas flow rate at standard temperature and pressure (cm^3^ (STP)/s), *l* is the membrane selective layer thickness (cm), *n* is the amount of fibers, ΔP is the different pressure between feed and system (cmHg) and A is effective surface area of membrane (cm^2^). The ideal selectivity, $\alpha_{i/j}$, of the membrane was calculated using Equation (2).
(2)$$\alpha_{i/j}=\frac{P_{i}}{P_{j}}$$
where P*_i_* is the permeability of gas *i*, and P*_j_* is the permeability of gas *j* (Barrer).

## 3. Results

### 3.1. Filler Preparation

The morphology of both zeolite-Y and ZCC are presented in Figure 1a,b. The Figure shows that zeolite-Y particles have hexagonal, diamond, rhombic and triangular crystal morphology. The smooth surface and sharp particle edge in zeolite-Y indicate that preparation using the gel method produced zeolite-Y with high crystallinity. The figure also shows that ZCC has similar particle morphology to zeolite-Y. This indicates that impregnation and the pyrolysis process did not break up the zeolite structure, which was consistent with the diffractogram data. However, the surface of ZCC was rougher, even though it still had sharp edges. The rough surface was coming from externally deposited sucrose. This is the reason why some of the peaks in the ZTC diffractogram had reduced intensity. Furthermore, the particle size distribution (PSD) of each sample is shown in d. The particle size distribution was measured using image J software based on the SEM image. The particle size distribution of zeolite-Y showed a sharp peak with no particles exceeding 1.5 µm. This indicates a very narrow particle size of the zeolite-Y formed. On the contrary, ZCC particles showed a broader distribution peak due to agglomeration formation, as can be seen in the SEM image. Even though ZCC had a broader particle distribution, the average particle size was slightly smaller compared to zeolite-Y, i.e., 0.53 ± 0.00021 compared to 0.59 ± 0.00061 µm, respectively. We assume that the pyrolysis at 800 °C may have shrunk the zeolite particles, which is somewhat similar to the sintering process in general.

The diffractogram of the zeolite-Y and ZCC are presented in Figure 1c which shows that the resulting zeolite-Y had high crystallinity. The peak observed at 2θ of ~6°, which corresponded to the (111) plane and which is usually known as the “basal peak”, was also present in the ZCC diffractogram data. This indicates the pyrolysis and sucrose impregnation did not destroy the zeolite pore structure [29]. The diffractogram of ZCC shows a similar peak pattern with the diffractogram of zeolite-Y. This indicates that the structure of zeolite-Y was preserved after the impregnation and pyrolysis process. However, there were some peaks that experienced some reduction in intensity. Those peaks were ~10° and ~32.8°, which correspond to the (220) and (840) planes, respectively. The reduced intensity may be caused by blockage by carbon deposits in those regions. Lower intensity of the diffractogram of ZCC than the zeolite-Y indicates that the crystallinity of ZCC may be reduced during the carbonization process. Nevertheless, based on the XRD analysis result, zeolite pore filling with carbon was successful without altering the zeolite structural framework.

### 3.2. Mixed-Matrix Membrane Preparation

The alteration in functional groups in P84 co-polyimide after ZCC incorporation was studied using Fourier transform infrared (FTIR) and illustrated in Figure 2. It can be noticed that all the membranes showed specific absorptions that appeared at wavenumbers of 720, 1360, 1715 and 1780 cm^−1^. The absorption that appeared at 720 cm^−1^ correspond to the C=O bond from the P84 co-polyimide precursor. The band at 1350 cm^−1^ correspond to the C–N, while the band at 1715 and 1780 correspond to the symmetric and asymmetric C=O, respectively. On the other hand, the ZCC spectra only showed a broad peak at ~1000 cm^−1^. This peak was a specific peak of zeolite-Y [37]. The FTIR spectra indicated that the main structure of zeolite-Y was not changed after sucrose impregnation and carbonization and agreed with the XRD and SEM data. Moreover, there was no new peak observed in the membrane after ZCC incorporation, indicating the interaction between P84 co-polyimide, and ZCC was physical, which is consistent with a report by Ebadi et al. [37].

Figure 3 shows diffractograms of the prepared membrane. The diffractograms of all membranes showed typical amorphous structures. The broad peaks that appeared at 10° and 35° 2θ corresponded to the amorphous structure of P84 co-polyimide membrane, which is in agreement with previously reported work [2]. With the introduction of ZCC into the polymer matrix, the amorphous peaks were decreased significantly in term of intensity. This confirms that the primary semi-crystalline internal structure of the polymer was being changed into a more rigid phase through the addition of ZCC [38]. A strong indication of the internal structure can typically be shown by the alteration in the *d*-spacing value between the Neat and MMM (P84/ZCC) membrane. However, due to very low amount of ZCC, the *d*-spacing was unobservable. With such *d*-spacing characteristics, the MMM is expected to have permeability improvement due to the increasing free volume [39]. A small peak appeared at 2θ~6°, corresponding to the zeolite-Y peak in ZCC structure. However, the other ZCC peaks of lower intensity from 2θ~6° were not observed due to very low amounts of ZCC being added. This is in agreement with the result from Dai et al. [40], namely that they were unable to observe the specific peaks in the diffractogram pattern of 13 wt% ZIF added into the Ultem polymer.

The filler distribution on the membrane surface and cross-section of the ZCC filled membrane were observed using SEM, as can be seen in Figure 4. The ZCC filled membrane showed a smooth surface and ideal filler–polymer interface. There was no obvious defect observed on the membrane nor the void between the filler and polymer interface. This indicates that ZCC had good compatibility toward P84 co-polyimide. The ZCC particles were dispersed homogenously on the membrane surface. The dense selective layer became thicker, i.e., from 0.83 µm to 1.79 µm, after the incorporation of ZCC. This might be attributed to the increase of dope solution viscosity at higher filler loading. Overall, the addition of ZCC into P84 co-polyimide produced a good particle–polymer interface with the polymer matrix with no interfacial gap observed visually. Such properties are essential in preparing composite membranes, since the presence of either void, surface rigidity or partial pore blockage by filler would greatly affect the gas separation performance. Naturally attached filler, usually by physical interaction, is preferred, since both polymer and filler will give the best contribution to the gas separation performance [41]. The membrane surface was getting smoother after the addition of ZCC, which was in agreement with the XRD data. The polymer chain was experiencing the rearrangement after the introduction of filler due to the adhesion force between filler and polymer body, which was indicated by the reduction of amorphous phase intensity (2θ~15°) on the XRD pattern. Moreover, the existence of the concentric cavities in the membrane indicated that there was a strong interaction between polymer and filler [42].

Thermogravimetric (TGA) and differential thermal analysis (DTA) were carried out to evaluate the thermal stability of the MMMs. The TGA and DTA curves of ZCC filled MMMs are presented in Figure 5. The degradation of all the prepared membrane occurred in two steps. The first was the evaporation of excess water at 100 °C causing around 5 wt% mass loss in each membrane, while the second was the starting decomposition of P84 co-polyimide that was different in each sample. The pristine membrane started to decompose at a temperature of 531.33 °C, which is consistent with the previously reported data [34,35,43]. This indicated the high thermal stability of P84 co-polyimide. As expected from previous work, the thermal stability of P84 membrane improved after the introduction of ZCC filler. The thermal stability was improved to 570.5 °C with the addition of 1 wt% of ZCC loading. Generally, the inorganic filler acts as heat absorber before being transferred to the membrane surface [39]. Thus, having well-dispersed filler on the membrane surface can greatly boost the thermal resistance of the membrane due to homogenous heat distribution on the membrane surface.

The glass transition temperatures (T_g_) of all prepared membranes were determined by differential scanning calorimetry (DSC). Generally, filler particles give a plasticization effect to the polymer membrane that reduce the T_g_ value [44]. The improved T_g_ value from 315 to 321 °C after 1 wt% of ZCC introduction means that there was macromolecular chain rigidification that occurred in the polymer. This could affect the gas transport properties of the membrane, with an improvement in selectivity [45].

### 3.3. Single Gas Permeation Test

The separation performance of the MMMs was evaluated to know the optimum loading of each filler. Five fibers of 15 cm in length were potted in a Swagelok 3164B3 and sealed with resin:hardener (2:1) prior to the measurement [36]. The single gas measurement was conducted at room temperature (~25 °C) and 2 bar feed pressure. The permeate side was connected to the bubble flow meter where the volumetric flow was acquired. Table 1 summarizes the single gas performance of the studied MMMs. For the uncoated PDMS Neat membrane, the O_2_ permeability was 7.12 Barrer, which was much higher compared to the literature [34,46,47]. The addition of ZCC improved the O_2_ permeability from 7.12 to 18.90 Barrer (2.65 times improvement). A minor defect on the membrane surface, which was created during the potting process, was expected. Consequently, the gas could freely move into the nonselective region originated from some defects on the surface, which led to high gas permeability but with low selectivity.

Simple treatment for resolving this issue was conducted by employing simple coating of highly permeable film on the membrane surface, such as PDMS [36,48]. As expected, the coated PDMS Neat membrane showed O_2_ permeability of 1.64 Barrer, which is consistent with the previously reported data [34,46,47,49]. The O_2_/N_2_ selectivity was improved from 4.11 to 7.32. The result indicated that the PDMS coating efficiently sealed the pin holes on the membrane surface and revealed the real intrinsic properties of P84 co-polyimide membrane. The separation performance was further improved after the introduction of ZCC particles in the membrane matrix. As expected from XRD results, the increment in *d*-spacing value resulted in increased permeability in the MMM. The O_2_ and N_2_ permeability of the coated MMMs was improved from 1.64 to 3.55 and 0.22 to 0.45 Barrer, respectively. This was due to the expansion of the polymer matrix from the introduction of fillers. The expansion of the polymer matrix results in increasing free volume inside the membrane. The free volume acts as a gas pathway through the membrane. A bigger gas pathway makes the gas penetrate the membrane easier, which agrees with the XRD data [39].

The selectivity of the ZCC loaded membrane increased slightly from 7.32 to 7.88 (average value is 7.65%). It was noted that the introduction of ZCC was able to improve O_2_ permeability by 216.46%, while preventing the selectivity drop. PDMS coated P84/ZCC membrane further improved the selectivity of O_2_/N_2_ separation from 4.92 to 7.88, which was slightly more than a 60% increase. The introduction of zeolite in the membrane usually leads to performance trade-off of the filled membrane compared to the respective Neat membrane [50]. However, such a phenomenon did not occur in this study. It is suspected that the carbon coating on the zeolite pores plays an important role in controlling the gas transport. First of all, the ZCC can be obtained during the zeolite templated carbon synthesis prior to zeolite removal using a strong acid like hydrogen fluoride (HF) and hydrogen chloride (HCl) [29]. At a glance, it is true that the material seems impossible to be applied as the filler due to a very low surface area, which is not effective for gas transport. However, we later found that the carbon that filled the zeolite pore was also a porous material. The pore size distribution of ZCC (0.73 ± 0.04 nm) was smaller compared to that of the zeolite-Y (0.86 ± 0.07 nm), in which the small pores prevented the selectivity drop of the membrane [29]. This means that the carbon did not block the pores of the zeolite. Instead, it covered the zeolite pores, thus increasing the microporosity, which contributed to the selectivity improvement. Moreover, the presence of the carbon layer prevented moisture from physically attaching to the zeolite pore through hydrogen bonding, and thus permeability reduction could be avoided. With this condition, water molecules in the ZCC could be easily removed even at low temperature (less than 100 °C). The significant permeability improvement was due to the pore regularity of ZCC and expansion of the free volume of the membrane. Moreover, the PDMS coating greatly affected the membrane permeability and selectivity performance in an equal manner. From a permeability point of view, the N_2_ reduction was greater than the reduction of O_2_, especially in the composite membrane. This might be attributed to the soft void formation in the polymer–filler interface, even though it was not observable in the SEM. Consequently, it improved the membrane selectivity. Overall, it can be concluded that the PDMS coating can be employed to observe the formation of soft defects on the membrane surface.

A comparison study was conducted to determine the significance of the study in the recent mixed matrix membrane progress. The studied membrane had higher selectivity than those of the previously reported results, as shown in Table 2. P84 co-polyimide is widely known to have decent gas separation performance in almost all gas pairs [51]. The performance was further improved with the addition of ZCC, even at low loading. Low loading of filler was preferred, as it avoids aggregate formation and increases the brittleness of the membrane. Moreover, as the inorganic filler preparation is costly, the amount of inorganic filler added into the polymer matrix should be considered carefully in order to produce economically viable mixed matrix membranes. The addition of ZCC into P84 co-polyimide matrix was promising to improve P84 co-polyimide separation performance.

The obtained permeabilities and ideal selectivities were inserted into a semi-empirical plot devised by Robeson [11,12]. Figure 6 illustrates the MMM performance in the Robeson curve for O_2_/N_2_ selectivity in comparison with data from some of the literature in Table 2. The PDMS coated membrane showed excellent O_2_/N_2_ separation, which is very good for medical application [14]. The P84/ZCC membrane nearly transcended the 2008 upper bound for polymeric performance, while the PDMS coated Neat membrane lied in the 1991 line. In addition, the uncoated ZCC membrane came near to the O_2_/N_2_ 1991 upper bound, which is also promising for the respective application. It was noted that all of the PDMS coated membranes belonged to the commercially attractive region [52]. A simple PDMS coating on the membrane was an effective method to recover the membrane performance due to minor surface defects. The membrane performance was further improved by actively controlling the gas transport through the membrane by inorganic filler addition.

## 4. Conclusions

In this study, simple zeolite modification by sucrose impregnation was successfully conducted. The obtained zeolite–carbon composite was able to reduce the particle size of the zeolite with a similar morphology structure. The FTIR results showed that the ZCC physically interacted with P84 co-polyimide. The ZCC particle distribution was well dispersed on the membrane surface and showed an ideal particle–polymer interaction with no gap/void observed. However, it was later confirmed that PDMS treatment allowed soft interfacial gaps to be detected between the polymer and fillers. Moreover, thermal resistance was increased by 7.37% after incorporation of 1 wt% of ZCC. The membrane performance of Neat membrane was maintained after PDMS coating with increased O_2_/N_2_ selectivity from 4.11 to 7.32. The membrane performance was further improved after ZCC loading with increased O_2_ permeability and O_2_/N_2_ selectivity by 216.46% and 7.65%, respectively. All of the PDMS coated membrane lies in the commercially attractive region for O_2_/N_2_ separation, which indicates that it is a promising candidate in air separation processes. Overall, the results suggest that the ZCC loading and PDMS coating were effective for improving the membrane performance.

## Figures and Tables

**Figure 1 membranes-10-00267-f001:**
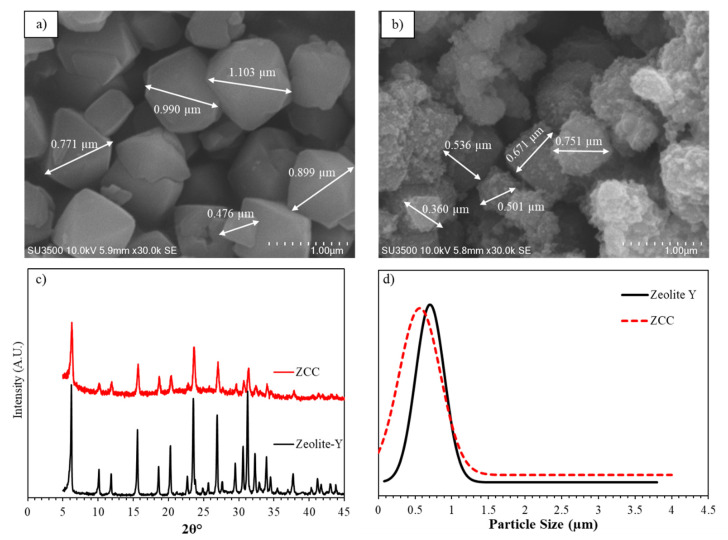
SEM images of (**a**) zeolite-Y and (**b**) zeolite carbon composite (ZCC) SEM; (**c**) X-ray diffractograms of zeolite-Y and ZCC; (**d**) particle size distribution of zeolite-Y and ZCC.

**Figure 2 membranes-10-00267-f002:**
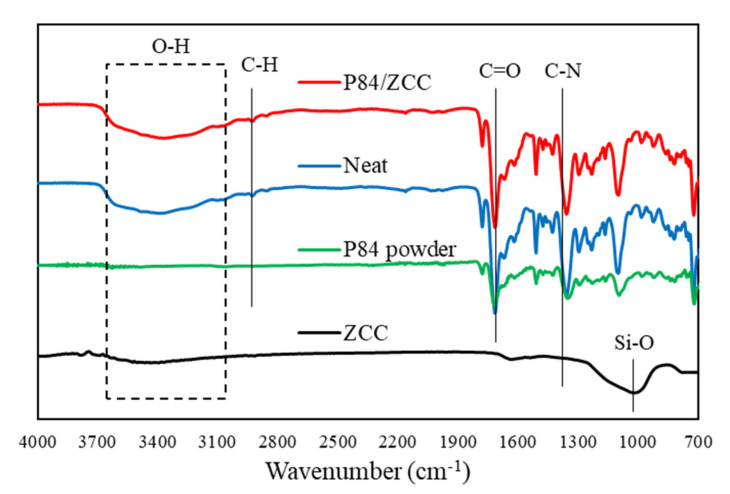
The FTIR spectra of all prepared samples.

**Figure 3 membranes-10-00267-f003:**
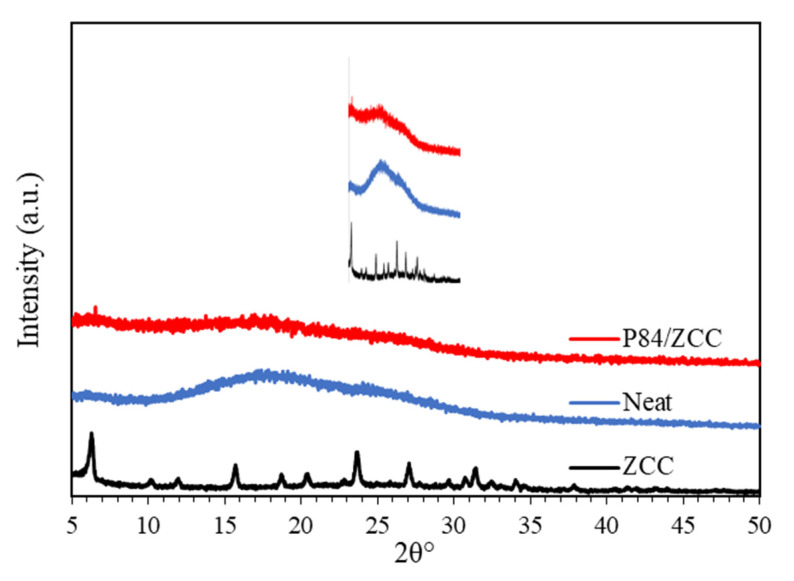
The X-ray diffractograms of prepared membranes.

**Figure 4 membranes-10-00267-f004:**
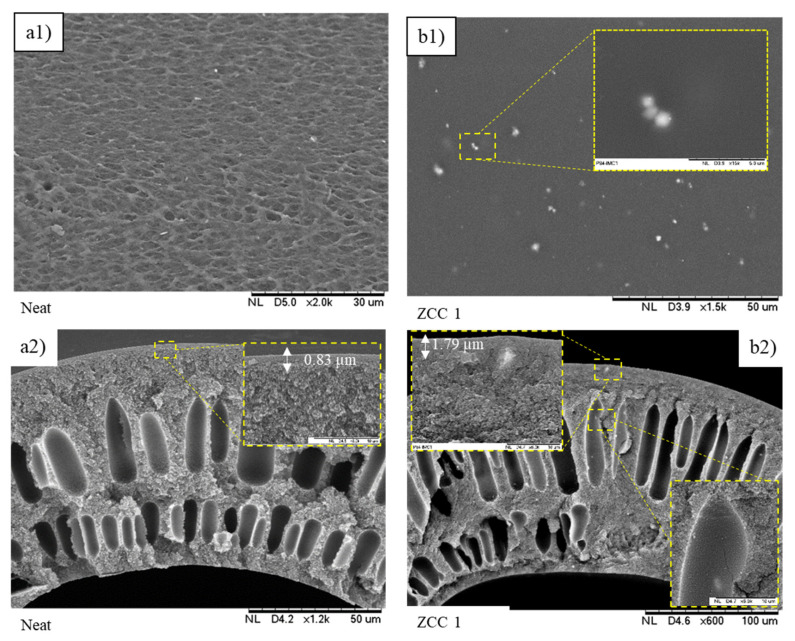
SEM image of surface and cross-section of (**a**) Neat and (**b**) P84/ZCC 1 membranes.

**Figure 5 membranes-10-00267-f005:**
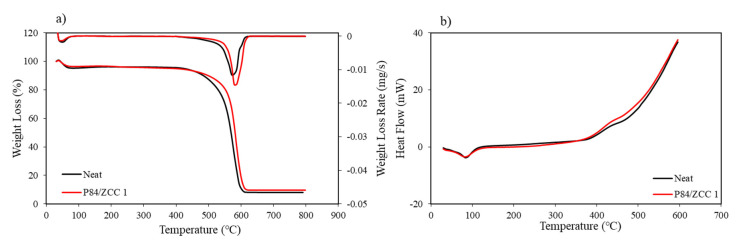
TGA (**a**) and DSC (**b**) curve of all prepared membranes.

**Figure 6 membranes-10-00267-f006:**
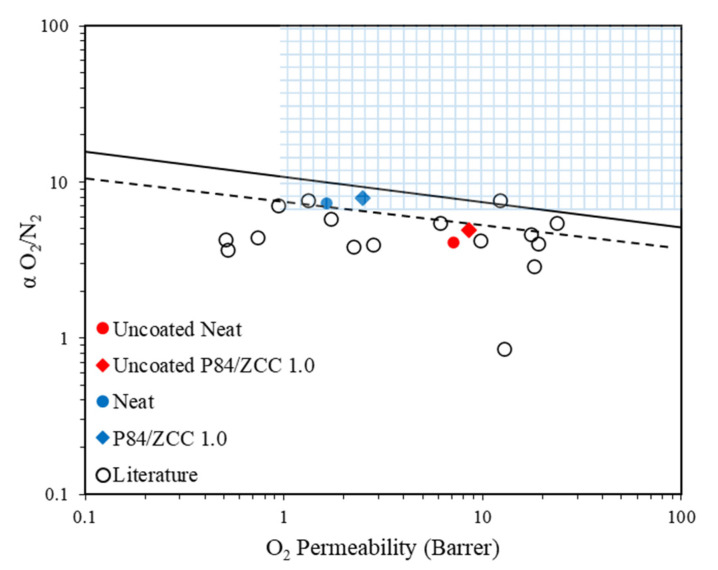
Performance of uncoated PDMS (red), coated membrane (blue) and other membranes from the literature with respect to the Robeson upper bounds.

**Table 1 membranes-10-00267-t001:** Single gas performance of mixed matrix membranes (MMMs).

Sample	Loading (wt%)	Permeability (Barrer)		Ideal Selectivity
		N_2_ 3.64 Å ^a^	O_2_ 3.46 Å ^a^	O_2_/N_2_
Uncoated PDMS				
Neat	0	1.73 ± 0.03	7.12 ± 0.20	4.11 ± 0.06
P84/ZCC1	1	3.84 ± 0.04	18.90 ± 0.51	4.92 ± 0.30
Coated PDMS 3 wt%				
Neat	0	0.22 ± 0.00 (−87.30%) ^b^	1.64 ± 0.03 (−76.96%) ^b^	7.32 ± 0.19 (+78.10%) ^b^
P84/ZCC1	1	0.45 ± 0.00 (−88.30%) ^b^	3.55 ± 0.00 (−81.20%) ^b^	7.88 ± 0.03 (+60.16%) ^b^

^a^ = kinetic diameter of gas (Å), ^b^ = number in the bracket indicate the improvement (+) and reduction (−) to the respective uncoated membrane.

**Table 2 membranes-10-00267-t002:** The O_2_/N_2_ separation performance of MMMs studied here as compared to the previously reported study.

Membrane	Filler Loading (wt%)	*p*O_2_ (Barrer)	αO_2_/N_2_	Ref.
P84	0	1.64	7.32	This work
P84/ZCC	1	3.55	7.88	
P84	0	2.8	0.9	[34]
Matrimid	0	1.72	5.79	[52]
Matrimid/CoPCMP	5	1.32	7.62	
Matrimid/Pluronic	5	0.93	7.09	
Pebax/T(p-OCH3)PPCoCl	0.6	12.2	7.6	[53]
PES	0	0.52	3.71	[21]
PES/Zeolite 13X	42	0.51	4.29	
PES/Zeolite 4A	42	0.74	4.40	
PU/Zeolite 3A	18	23.93	2.18	[22]
PU/ZSM-5	18	25.21	2.45	
PSF/CNF	1	2.24	3.86	[54]
PU	0	2.8	4	[55]
PU/UiO-66 (Zr)	28	6.1	5.5	
PU/MIL-101 (Cr)	28	9.7	4.2	
PES-Matrimid/Zeolite 4A	30	12.8	0.85	[23]
ODPA-TMPDA	0	17.5	4.62	[56]
ODPA-TMPDA/SNW	10	23.6	5.44	

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
