# Peer review of "P84/ZCC Hollow Fiber Mixed Matrix Membrane with PDMS Coating to Enhance Air Separation Performance"

_membranes, 2020, doi:10.3390/membranes10100267_

Round 1

Reviewer 1 Report

The research is very interesting and has potential for industrial application. But I have some suggestions and questions:

1) In the FTIR results (lines 195 - 197), the authors say that the interaction between P84 co-polyimide and ZCC was physical. I think that with only 1% of ZCC incorporated it will be impossible to see any band on FTIR. So I think that the authors cannot make this conclusion.

2) In the XRD results: Lines 203 - 204, occurs the same, the quantity of ZCC (1%) are very few to see any modification on XRD. I think that e cannot say that are some kind of order because of the lost of amorphous background.

3) In the XRD results: lines 207 - 209, I think that is not correct measure d-spacings in a so broad peak. (in the amorphous band).

4) The results of the Permeation test prove that the material with 1% of ZCC is better than without it. But it is not clear why this occurs.

5) First of all, why a Zeolite with the porous completed with carbon would be more effective for the separation of gases? That is not logical.

6)Textural measurements for Area and pores volumes must be done.

7) Why more ZCC content don't lead to a more effective separation?

Author Response

Responses to the Reviewer Comments
Manuscript ID: membranes-901197

Reviewer 1

The research is very interesting and has potential for industrial application. But I have some suggestions and questions:

1) In the FTIR results (lines 195 - 197), the authors say that the interaction between P84 co-polyimide and ZCC was physical. I think that with only 1% of ZCC incorporated it will be impossible to see any band on FTIR. So I think that the authors cannot make this conclusion.

Answer:

Thank you for the question. The reason behind our hypothesis is based on previous result which was reported by (Ebadi et al., 2015) who incorporated NaY (15 wt%) into Matrimid matrix and showing no chemical interaction between the membrane and zeolit NaY. (Line 210)

2) In the XRD results: Lines 203 - 204, occurs the same, the quantity of ZCC (1%) are very few to see any modification on XRD. I think that we cannot say there are some kind of order because of the lost of amorphous background.

3) In the XRD results: lines 207 - 209, I think that is not correct measure d-spacings in a so broad peak. (in the amorphous band).

Answer to comments 2) and 3):

The decrease in the amorphous region of P84 due to the addition of filler was also observed previously (Baneshi et al., 2020; Ebadi et al., 2015; Hua et al., 2014) which suggest the fillers have been successfully incorporated into the P84 matrix and resulting in the alteration of internal primary crystalline structure of the P84. However, we agree with reviewer comment that 1% ZCC is to small to be observed by XRD. Therefore, we made some revision on the discussion of Figure 3 (diffractograms of P84/ZCC, Neat and ZCC). A small satelite picture is added to Figure 3 to show that the d-spacing alteration is not observed.

“Figure 3 shows diffractograms of the prepared membrane. The diffractogram of all membrane shows typical amorphous structure. The broad peak that appeared at 2θ of 10-35 was correspond to the amorphous structure of P84 co-polyimide membrane, in agreement with previously reported work [2]. With the introduction of ZCC into polymer matrix, this amorphous peak was decreased significantly in term of intensity. This confirms that the primary semi-crystalline internal structure of the polymer was being changed into a more rigid phase throughout the addition of ZCC [37]. Strong indication on the internal structure shall be shown by the alteration in d-spacing value between Neat and MMMs (P84/ZCC) membrane. However, due to very low amoun of ZCC, the d-spacing was unobservable. With such d-spacing characteristic, the MMMs membrane is expected to have permeability improvement due to the increasing free volume [38]. A small peak appeared at 2θ~6° corresponds to the zeolite-Y peak in ZCC structure. However, the other ZCC peaks that has lower intensity from 2θ~6° was not observed due to very low amount of ZCC were added. This is in agreement with the result from Dai et al. [39] that unable to observe the specific peaks of 13 wt% ZIF added into Ultem polymer in diffractogram pattern”. This revised paragraph was written in line 213-228.

4) The results of the Permeation test prove that the material with 1% of ZCC is better than without it. But it is not clear why this occurs.

5) First of all, why a Zeolite with the porous completed with carbon would be more effective for the separation of gases? That is not logical.

Answer for question 4 and 5:

First of all the ZCC can be obtained during the zeolite templated carbon synthesis prior to the zeolite removal using strong acid like HF and HCl (Gunawan et al., 2018). At glance it is true that the material seems impossible to be applied as the filler due to very low surface area which is not effective for gas transport. However, we later found that the carbon that filled the zeolite pore is also a porous material. The pore size distribution of ZCC (0.73±0.04 nm) is smaller as compared to the zeolite-Y (0.86 ± 0.07 nm) (Gunawan et al., 2018). This means that the carbon is not blocking the pore of the zeolite, instead it is covering the zeolite pores, thus increasing in the microporosity which contribute to the selectivity improvement. Moreover, the presence of carbon layer prevent moisture to physically attached to the zeolite pore through the hydrogen bonding, thus the permeability reduction can be avoided.

6)Textural measurements for Area and pores volumes must be done.

Answer:

The surface characteristic of ZCC in this study was already reported in our previous paper (Gunawan et al., 2018).

7) Why more ZCC content don't lead to a more effective separation?

Answer:

Usually, more filler added will result in the void formation on the polymer-filler interface which result in the selectivity loss. Moreover, large amount of filler added into polymer matrix would change the membrane flexibility that lead to the brittle properties.

Reviewer 2 Report

In this manuscript, the authors investigated the effect of zeolite-carbon composite fillers on the O2/N2 separation performance of P84 hollow fiber membranes. The manuscript is well written and the conclusions are generally supported by the characterization data. I do feel that the manuscript could be published in Membranes after addressing the following relatively minor issues:

1. The manuscript title emphasizes the enhanced air separation performance of membranes, however, the presented gas data are only based on the single gas separation efficiency of O2 and N2. I would recommend authors to test the separation performance of their membranes under air (79% N2-21% O2) rather than pure O2 and N2 gases.

2. In Figure 1, SEM images, it seems that the average size of zeolites is slightly larger than the composite fillers. Although the particle size distributions have been reported (Figure 1(d)) and are in the same range. It is not clear what has been the method for PSD measurements.

3. I was wondering why the authors reported their gas data based on barrer (permeability) rather than permeance (GPU). How the authors calculated the membrane selective layer thickness here. Please include the thickness data in the manuscript.

4. On Page 2, line 66, STP usually refers to (0℃ and 1 atm) not ambient condition.

5. In table 2, authors have compared their separation performance with some other reported literature data. To rationalize the concept of this work, i.e. using carbon-coated zeolites as filler to develop mixed matrix membranes, authors must provide some gas data of P84/neat zeolite as the control sample. This could be presented according to the available literature.

Author Response

Response to the Reviewer 2 can be seen in the attachment. Thank you

Round 2

Reviewer 1 Report

Thank you for your responses. Now the results and the study is more clear to me.

I think that the answer to questions 4 +5 and 7 must be included in the text to be more clear to the readers.

Author Response

Dear Reviewer,

Please see the attachment. Thank you

Best regards,

Nurul Widiastuti
